# Application of optogenetic Amyloid-β distinguishes between metabolic and physical damages in neurodegeneration

Chu Hsien Lim[1†], Prameet Kaur[1†], Emelyne Teo[1], Vanessa Yuk Man Lam[1], Fangchen Zhu[1], Caroline Kibat[1,2,3], Jan Gruber[1,4], Ajay S Mathuru[1,2,3], Nicholas S Tolwinski[1]*

[1]Science Division, Yale- NUS College, Singapore, Singapore; [2]Institute of Molecular and Cell Biology (IMCB), Singapore, Singapore; [3]Department of Physiology, YLL School of Medicine, Singapore, Singapore; [4]Department of Biochemistry, National University of Singapore, Singapore, Singapore

**Abstract** The brains of Alzheimer's disease patients show a decrease in brain mass and a preponderance of extracellular Amyloid-β plaques. These plaques are formed by aggregation of polypeptides that are derived from the Amyloid Precursor Protein (APP). Amyloid-β plaques are thought to play either a direct or an indirect role in disease progression, however the exact role of aggregation and plaque formation in the aetiology of Alzheimer's disease (AD) is subject to debate as the biological effects of soluble and aggregated Amyloid-β peptides are difficult to separate in vivo. To investigate the consequences of formation of Amyloid-β oligomers in living tissues, we developed a fluorescently tagged, optogenetic Amyloid-β peptide that oligomerizes rapidly in the presence of blue light. We applied this system to the crucial question of how intracellular Amyloid-β oligomers underlie the pathologies of A. We use *Drosophila*, *C. elegans* and *D. rerio* to show that, although both expression and induced oligomerization of Amyloid-β were detrimental to lifespan and healthspan, we were able to separate the metabolic and physical damage caused by light-induced Amyloid-β oligomerization from Amyloid-β expression alone. The physical damage caused by Amyloid-β oligomers also recapitulated the catastrophic tissue loss that is a hallmark of late AD. We show that the lifespan deficit induced by Amyloid-β oligomers was reduced with Li[+] treatment. Our results present the first model to separate different aspects of disease progression.

*For correspondence:
nicholas.tolwinski@yale-nus.edu.sg

†These authors contributed equally to this work

**Competing interests:** The authors declare that no competing interests exist.

## Introduction

Alzheimer's disease (AD) is a debilitating, age-associated, neurodegenerative disease which affects more than 46.8 million people worldwide and represents the 6th leading cause of death in the United States of America (*Ahn et al., 2001*; *De-Paula et al., 2012*; *Hawkes, 2016*; *Kumar et al., 2015*; *Zhang et al., 2011*). Despite extensive efforts over the last 50 years, no disease-modifying therapy has been found and, most recently, several high-profile Phase III clinical trials have failed (*Anderson et al., 2017*; *Lim and Mathuru, 2018*; *Park et al., 2018*). The AD clinical trial landscape has largely been dominated by the amyloid cascade hypothesis, with more than 50% of the drugs targeting amyloid beta (Aβ) in Phase III trials alone (*Cummings et al., 2018*). The amyloid cascade hypothesis, in its original form, posits the deposition of Aβ, in particular the extracellular Aβ plaque, as the main driver of AD (*Hardy and Higgins, 1992*). However, the causative role of Aβ plaques has recently been challenged, because of the failure of numerous interventions targeting Aβ plaques in Phase III trials and the observation of Aβ plaques in brains of non-AD symptomatic individuals (*Cummings, 2018*). Model organisms ranging from nematodes to mice have further shown that AD pathology can be modeled in the absence of obvious Aβ plaques (*Duff et al., 1996*; *Fong et al.,*

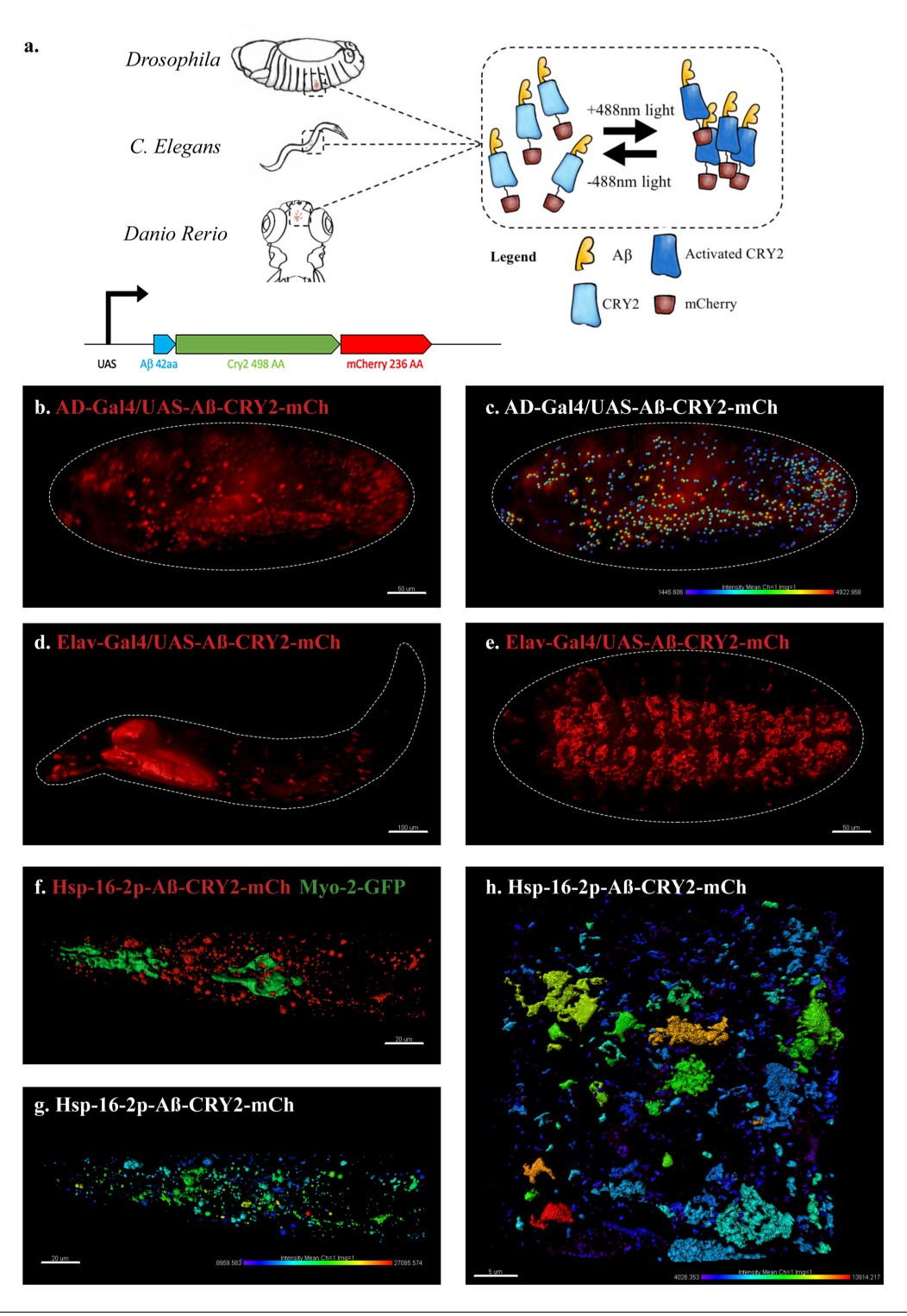

**Figure 1.** An in vivo, light-dependent, oligomerization switch for the formation or dissolution of Aβ aggregates in the fruit fly and nematode. (a) A schematic of the strategy and a relative size comparison of the three components of UAS-Aβ-CRY2-mCh. (b) Expression of AD-Gal4/UAS-Aβ-CRY2-mCh in a *Drosophila* embryo. (c) Mean intensity of aggregates in the same AD-Gal4/UAS-Aβ-CRY2-mCh*Drosophila*embryo. (d and e) Expression of Elav-Gal4/UAS-Aβ-CRY2-mCh in a *Drosophila* larva and embryo. (f) Expression of hsp-16–2 p-Aβ-CRY2-mCh in heat-shocked *C. elegans* with *myo-2-gfp* marker, that marks its pharyngeal pump and serves as an indicator for positive

*Figure 1 continued on next page*

*Figure 1 continued*

microinjection. (**g and h**) Mean intensity of aggregates in the hsp-16–2 p-Aβ-CRY2-mCh in *C.elegans* under 20x and 63x magnification of the confocal microscope; 63x image was processed using the Zeiss Airyscan.

The online version of this article includes the following figure supplement(s) for figure 1:

**Figure supplement 1.** Transgenic *Drosophila* model expressing Aβ-CRY2-mCh.

**Figure supplement 2.** Exposure to light drives Aβ aggregation in *C. elegans* strains 24 hr post heat- shock.

**Figure supplement 3.** Light-induced Aβ aggregation in *Drosophila* embryos resulted in distinct developmental defects.

**Figure supplement 4.** Thioflavin T (ThT) staining showing the Aβ aggregation transgenic *Drosophila* embryos ubiquitously expressing Aβ-CRY2-mCh.

---

*2016*; *Teo et al., 2019*). These observations point to a different mechanism for Aβ neurotoxicity perhaps through Aβ's intracellular accumulation (*LaFerla et al., 2007*).

Amyloid plaques are macroscopic extracellular protein aggregates, but intracellular Aβ must function in smaller units of peptides, oligomers or aggregates. Of these, soluble Aβ oligomers appear to be the main toxic species in AD (*Ferreira and Klein, 2011*). For example, Aβ oligomers have been shown to induce neurotoxic effects including memory loss in transgenic animal models reviewed in *Mroczko et al. (2018)*. These studies, however, mostly rely on in vitro injection of synthetic Aβ oligomers or oligomers extracted from AD brains (*Mroczko et al., 2018*). There is currently a lack of tools that can directly control Aβ oligomerization in vivo, which would allow direct examination of the effects of Aβ oligomerization.

Here, we describe an optogenetic method to study Amyloid-β (Aβ) protein oligomerization in vivo in different model organisms and apply it to delineating mechanisms of disease progression. Optogenetics hinges on light responsive proteins that can be fused to genes of interest, allowing for the spatial and temporal regulation of proteins in a highly precise manner. Temporal-spatial control is achieved simply by the exposure of the target system to lights of a specific wavelength, without the need to introduce other external agents (*Fenno et al., 2011*; *Möglich and Moffat, 2010*). One such optogenetic protein, a modified version of the *Arabidopsis thaliana* cryptochrome 2 (CRY2) protein, oligomerizes into photobodies quickly and reversibly in the presence of blue light at 488 nm (*Más et al., 2000*). In cultured cells, expression of the photolyase homology region of CRY2 fused to mCherry led to puncta within 10 s of exposure to blue light and these puncta dispersed within minutes (*Bugaj et al., 2013*). This basic unit of CRY2-mCherry attached to a variety of proteins have, for instance, been used to study cortical actin dynamics in cell contractility during tissue morphogenesis, the dynamics of Wnt and EGF signaling, plasma membrane composition and transcriptional regulation (*Bugaj et al., 2018*; *Bugaj et al., 2015*; *Guglielmi et al., 2015*; *Huang et al., 2017*; *Idevall-Hagren et al., 2012*; *Johnson et al., 2017*; *Kaur et al., 2017*; *Kennedy et al., 2010*).

## Results

To address the role of aggregation of intracellular Aβ in the pathophysiological effects of AD, we developed a light-inducible system for use in model organisms. Specifically, we generated an in vivo, light-dependent, oligomerization switch for the formation and dissolution of Aβ oligomers in *Drosophila melanogaster*, *Caenorhabditis elegans* and *Danio rerio* (*Figure 1a*). This optogenetically-driven model allowed the visualization of microscopic Aβ oligomerization and addressed the question of their relationship to Aβ pathogenesis with spatiotemporal precision.

### Generation of transgenic models with light-inducible intracellular Aβ oligomerization in model organisms

We generated transgenic animals expressing the 42-amino-acid human Aβ peptide (Aβ1–42) fused to Cryptochrome 2 and the fluorescent protein mCherry (Aβ-CRY2-mCh, *Figure 1a*, please note the relative sizes). *Drosophila* lines were under the control of the GAL4/UAS system (*Brand and Perrimon, 1993*) and expressed either ubiquitously via AD-Gal4 driver (*Figure 1b–c*) or specifically in neurons using Elav-Gal4 driver (*Figure 1d–e*, *Figure 1—figure supplement 1*). *C. elegans* lines were made with the heat shock promoter (*Figure 1f–h*, *Figure 1—figure supplement 2*). In order to observe oligomerization and clustering of intracellular Aβ, we took a live imaging approach in

*Drosophila* embryos using a lightsheet microscope with a 488 nm laser activating CRY2 and the 561 nm laser imaging mCh over time. We observed clusters of mCh fluorescence forming in embryos exposed to 488 nm light, but fewer in embryos that were not exposed to blue light (*Figure 1—figure supplement 3*, *Videos 1–2*). The intensity of the clusters was quantified using Imaris quantification software to show the intensity of oligomer formation (*Figure 1c*, note color scale for cluster intensity).

We proceeded to test the effectiveness of light induced clustering of Aβ-CRY2-mCh in *C. elegans*. As this construct is driven by a heat shock promoter, worms were heat shocked at 35℃ for 90 min followed by resting at 20℃ before being immobilized and imaged using confocal microscopy (*Figure 1f–h*). We observed an increase in clusters in worms exposed to blue light over their siblings not exposed to blue light. The results were quantified as fluorescence intensity and represented in (*Figure 1h*) showing the formation of Aβ clusters.

Induction of chaperones from heat-shock can impact the dynamics of Aβ. In particular, heat-shock treatment in another transgenic *C. elegans* strain, CL4176, reduced Aβ-mediated paralysis and Aβ oligomerization (*Winblad, 1989*; *Wu et al., 2010*). Overexpression of chaperone protein suppressed Aβ-toxicity in *C. elegans* (*Fonte et al., 2008*). The use of hsp-16 promoter in the transgenic *C. elegans*, which requires heat-shock induction, may hence be a limitation of our current approach. In view of this limitation, we performed several other controls to understand the impact of heat-shock treatment on Aβ dynamics, and whether or not heat-shock treatment itself contributes to the metabolic and phenotypic detriments observed in the transgenic animals.

To understand how heat-shock treatment may impact the dynamics of Aβ in our transgenic strains, we compared the dynamics of Aβ between the transgenic Aβ worms with and without heat shock. We found that transgenic animals in both heat-shock (Aβ HS L) and non-heat-shock (Aβ -HS L) condition expressed detectable Aβ-CRY2-mCh protein (*Figure 1—figure supplement 2*), however animals that had been heat-shocked had significantly higher Aβ levels compared to non-heat-shocked animals, suggesting that heat-shock, as intended, drove higher level of Aβ expression and that this was not compensated for by secondary induction of chaperones.

Even though the CRY2 system predominantly oligomerizes, we also observed the formation of intracellular Aβ aggregates, using a direct stain for Aβ aggregates (*Figure 1—figure supplement 4*). Importantly, we observed that Aβ aggregation was not reversible as had previously been observed for signaling molecules (*Huang et al., 2017*; *Johnson et al., 2017*; *Kaur et al., 2017*), rather CRY2 appeared to initiate clustering leading to Aβ bundles that did not come apart once blue light was turned off. We further confirmed that turning on blue light alone without the transgene expression did not lead to any Aβ expression or cluster formation (*Figure 1—figure supplement 2b*, Ab HS L vs Ab –HS L), suggesting that the oligomerization process was CRY2-specific and not an artifact of

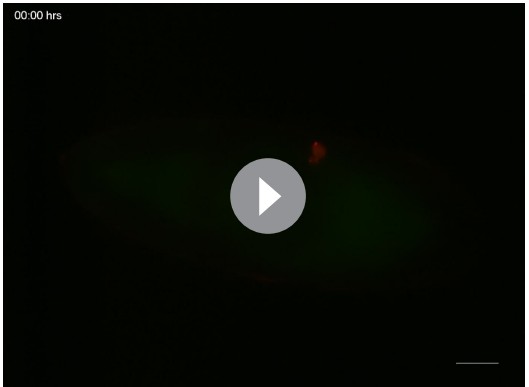

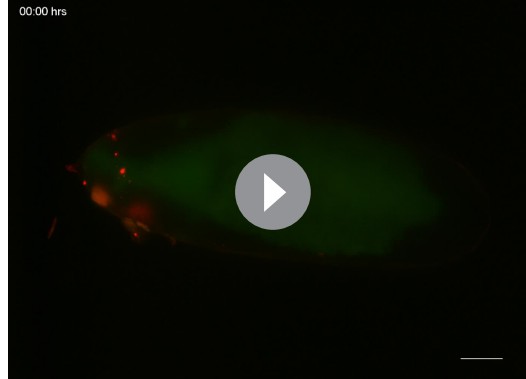

**Video 1.** Elav-Gal4/UAS-Aβ-CRY2-mCh embryos kept in the dark and imaged for neurons in red mCherry and glial cells Repo-QF2 >QUAS GFP. Blue laser power kept low to allow imaging of glial cells, but not high enough to activate aggregation.
https://elifesciences.org/articles/52589#video1

**Video 2.** Elav-Gal4/UAS-Aβ-CRY2-mCh embryos exposed to light and imaged for neurons in red mCherry and glial cells Repo-QF2 >QUAS GFP. Blue laser power at higher setting to allow imaging of glial cells and activate aggregation.
https://elifesciences.org/articles/52589#video2

light. We performed FRAP experiments to establish the stability of the Aβ clusters (*Figure 2*). When compared to an unrelated CRY2 construct, we observed a very slow recovery for Aβ-CRY2-mCh compared to Arm-CRY2-mCh (*Figure 2*; *Kaur et al., 2017*). Together, our results showed that the Aβ-CRY2-mCh transgene was functional in both organisms, with a high specificity to blue light inducing irreversible oligomerization of Aβ.

## Light-inducible Aβ oligomerization causes lifespan and behavioral deficits in transgenic models

To test the functionality of Aβ-CRY2-mCh, we next looked for phenotypes associated with light-induced intracellular Aβ oligomerization. We performed lifespan studies to assess the effects of Aβ-CRY2-mCh by expressing Aβ-CRY2-mCh either ubiquitously (AD-Gal4) or specifically in the nervous system (Elav-Gal4). Newly eclosed adult flies were separated into two groups in each case, one of which was exposed to ambient white light and the other kept in the dark. In both cases, flies exposed to light died much more quickly with half the mean and maximum lifespans (*Figure 3a–b*) compared to those reared in the dark. We extended this analysis to *C. elegans,* though starting at Day 6 due to the heat-shock manipulation step. Under light conditions, *C. elegans* showed markedly decreased lifespans as compared to the transgenic *C. elegans* kept in dark conditions (*Figure 3c*). Most significantly, we observed that the lifespan decrease was reversible, as taking animals exposed to light for 24 days and moving them into darkness showed a recovery of their lifespan, or a rescue (*Figure 3a–b*). To ensure that these reduced lifespans were physiological, we examined the fitness of the animals. Fitness was severely decreased in light-exposed flies and worms as quantified by fertility and locomotive assays (*Figure 3—figure supplement 1*). Interestingly, light-induced Aβ oligomerization caused further impairment in sensorimotor function of transgenic Aβ nematodes (*Figure 3—figure supplement 1c*). These results suggested that light-induced oligomerization of Aβ not only affected lifespan and fitness, but also impaired complex sensorimotor function and behavior.

## Light-inducible Aβ oligomerization leads to physical and metabolic damage in transgenic models

The lethality mediated by light-induced Aβ oligomerization led us to examine the morphological effects of Aβ-CRY2-mCh clustering during embryonic stages in *Drosophila*. Embryos expressing Aβ in only the nervous system developed at a normal pace and did not show any morphogenetic defects and even hatched in the absence of blue light (*Video 1*). In contrast, sibling embryos exposed to blue laser light arrested in late neurogenesis stages where the central nervous system stopped developing and the embryos died (*Video 2* and *Figure 4*, *Figure 1—figure supplement 3* and *Figure 4—figure supplement 1*). These embryos showed a variety of abnormalities in the positioning of neurons and glial cells (Compare to complete normal development, *Video 3*) suggesting that light-induced Aβ oligomerization caused physical, structural damage that affects neuronal development.

The severity of the physical damage phenotype in embryos exposed to blue light (*Figure 4a–d''*) led us to investigate the differences in mechanism between the two conditions by exploring inflammatory response, mitochondrial health and oxidative damage. Expression of Aβ alone, without light-induced Aβ oligomerization, led to a small increase in inflammatory markers (*Figure 4e*; *Westfall et al., 2018*). In embryos exposed to light, two of these markers increased to a much greater level (*Figure 4e*). We observed a significant reduction in the number of mitochondria in light-treated Aβ animals (*Figure 4f*). Markers of oxidative damage in light and dark exposed *C. elegans* showed differential expression of antioxidant defense genes, specifically Catalase, Trx-2 and Sod-3, suggestive of the occurrence of oxidative stress induced by Aβ oligomerization (*Figure 5*). These results suggested that light-induced Aβ oligomerization phenocopied many of the hallmarks of AD.

We proceeded to look at light-induced Aβ oligomerization in metabolic impairments in transgenic *C. elegans.* Heat-shocked mutants exposed to light (Aβ HS L) had lower ATP levels compared to the non-transgenic control (Ctrl -HS D) (*Figure 6*). However, heat-shocked mutants in the dark condition (Aβ HS D) did not show any significant differences in ATP level compared to the non-transgenic control. This result suggested that presence of Aβ alone is insufficient to induce ATP deficits, and that

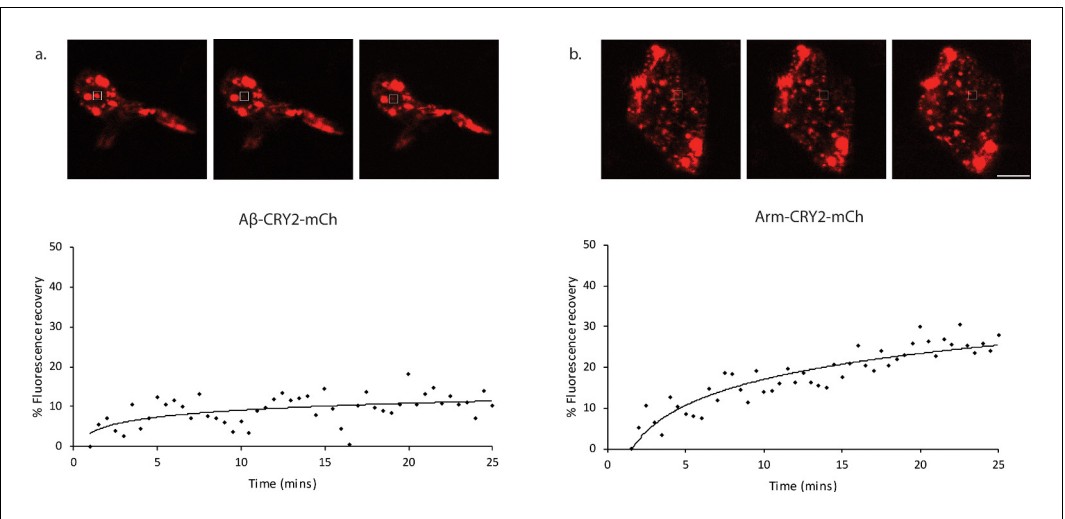

**Figure 2.** Fluorescence recovery after photobleaching (FRAP) analysis of Aβ-CRY2-mCh expressing cells compared to Arm-CRY2-mCh expressing cells in *Drosophila* gut. Images and percentage fluorescence recovery after bleaching of (a) Aβ-CRY2-mCh expressing intestinal stem cells from *esg-Gal4, UAS-GFP; UAS-Aβ-Cry2-mCh* flies and (b) Arm-CRY2-mCh expressing stem cells from *esg-Gal4, UAS-GFP; UAS-*Arm-*Cry2-mCh* flies. Aβ-Cry2-mCh showed a much slower recovery of fluorescence post-bleaching as compared to an unrelated CRY2 construct (Arm-CRY2-mCh). Scale bar represents 5 μm.

light-induced oligomerization of Aβ is required for the defect to manifest. Aβ expression alone was also insufficient to affect the nematodes' maximum respiratory capacity, as heat-shocked mutants in the dark condition (Aβ HS D) did not display differences in maximum and spare respiration capacity compared to control animals (Ctrl -HS D). However, light-induced Aβ oligomerization significantly reduced maximum respiration capacity in *C. elegans*. Heat-shocked mutants in the light condition (Aβ HS L) displayed significantly lower maximum and spare respiration capacity compared to controls (Ctrl -HS D) and to heat-shocked mutants in the dark condition (Aβ HS D) (*Figure 6b–d*). Together, these results highlighted the metabolic differences between the two conditions and showed that metabolic deficits were mediated by light-induced Aβ oligomerization.

To further examine whether the metabolic deficits in transgenic *C. elegans* were mediated by heat-shock treatment, we included a heat-shocked non-transgenic control in the metabolic and phenotypic assays. We found that heat-shock treatment does not affect ATP levels in the control animals, suggesting that the ATP detriment was a result of Aβ expression, but not heat-shock treatment (*Figure 6a*). On the other hand, heat-shock treatment appears to reduce the respiratory capacity of the control animals, suggesting that the decline in respiratory capacity was mediated by both heat-shock and Aβ expression (*Figure 6c*). Comparing the heat-shocked transgenic animals in light and dark conditions, we found that light treatment, which induces oligomerization, worsened the phenotypic and metabolic detriments in the transgenic animals. This suggested that regardless of the effects of heat-shock and chaperone induction, oligomerization still negatively impacted the phenotypes.

## Light-inducible Aβ lifespan decrease is rescued by Li[+]

The drug Li[+] is used for treatment of neurological conditions (*Licht, 2012*). A high drinking water Li[+] concentration correlated with lower human mortality (*Zarse et al., 2011*), and Li[+] increased the lifespan of *C. elegans* and *Drosophila* (*Castillo-Quan et al., 2016*; *McColl et al., 2008*; *Tam et al., 2014*; *Teo et al., 2020a*; *Teo et al., 2020b*; *Zarse et al., 2011*). The molecular mechanisms by which Li[+] functions include inhibition of inositol monophosphatase (IMPA) and glycogen synthase kinase-3 (GSK-3) (*Kerr et al., 2018*), reduction of oxidative damage (*Kasuya et al., 2009*; *Kerr et al., 2017*; *Khan et al., 2015*) and longevity via a GSK-3/NRF-2 dependent, hormetic mechanism (*Castillo-Quan et al., 2016*). Among the mechanisms for neuroprotection, the Wnt signaling pathway plays a role in modulating AD pathology and its progression (*De Ferrari et al., 2003*; *Jin et al., 2017*;

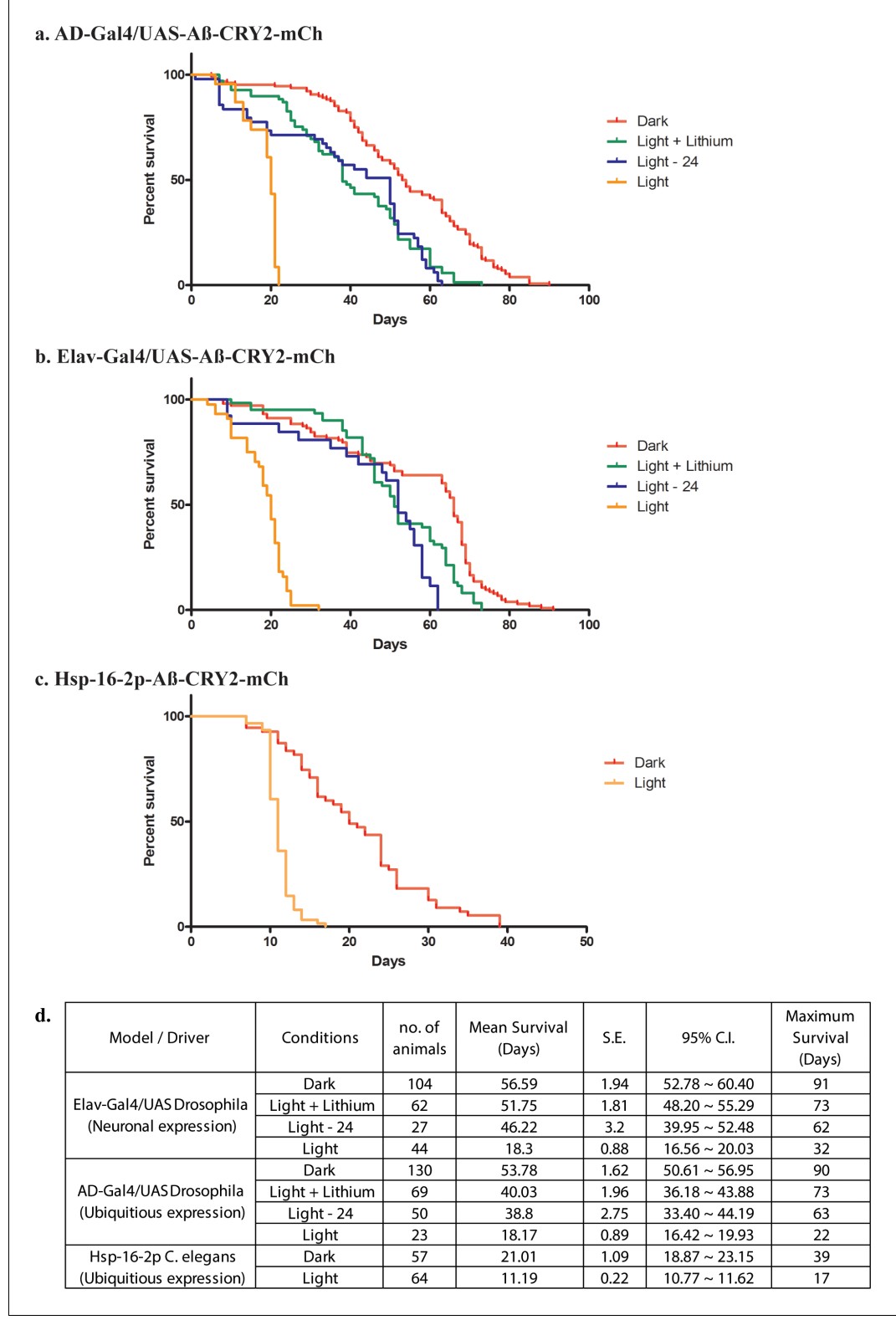

Figure 3. Analysis of lifespans in transgenic Aβ-aggregation *C. elegans* and *Drosophila* and adult models. (**a**) Kaplan-Meier survival curves of transgenic Aβ *Drosophila* adults driven by AD-Gal4 kept in the dark, exposed to light, exposed to light for 24 days then moved to the dark and exposed to light but fed with a lithium-supplemented diet. (**b**) Kaplan-Meier survival curves of transgenic Aβ *Drosophila* adults driven by Elav-Gal4 kept in the dark, exposed to light, exposed to light for 24 days then moved to the dark and exposed to light but fed with

*Figure 3 continued on next page*

*Figure 3 continued*

a lithium-supplemented diet. (**c**) Kaplan-Meier survival curves of control non-heat shocked nematodes, transgenic Aβ heat-shocked worms with the hsp-16–2 p promoter kept in the dark and transgenic Aβ heat-shocked worms with the hsp-16–2 p promoter kept in the light; those in light show a significantly shortened lifespan (overall significance by log-rank test: ****p<0.0001) (**d**) Table showing number of samples used with the mean and maximum survival times for all conditions.

The online version of this article includes the following figure supplement(s) for figure 3:

**Figure supplement 1.** Aβ-CRY2-mCh expressing transgenic *C. elegans* and *Drosophila* reared in light showed signs of reduced fitness.

---

*Parr et al., 2015*; *Toledo and Inestrosa, 2009*). Activation of Wnt signaling through lithium (Li⁺) treatment attenuated Aβ aggregation and increased the lifespan of Aβ models (*Toledo and Ines-trosa, 2009*). By inhibiting the enzyme glycogen synthase kinase-3 (GSK-3), Li⁺ compounds drive the downstream production of intracellular β-catenin activating the Wnt–β-catenin signaling pathway (*Klein and Melton, 1996*). Li⁺ treatment through Wnt activation also rescued behavioral impairment and neurodegeneration induced by Aβ fibrils (*De Ferrari et al., 2003*). Taken together, these studies point to a putative AD therapeutic intervention through Li⁺. To test the usefulness of optogenetic Aβ for drug testing, we supplemented the food for *Drosophila* expressing Aβ-CRY2-mCh with Li⁺. *Drosophila* exposed to light, but consuming Li⁺ had significantly increased lifespans, suggesting both a rescue effect induced by Li⁺ (*Figure 3a–b*), and that the optogenetic model can be used for inducible expression drug testing. Rapid drug testing can be also be achieved through cell culture. We extended this assay to HEK293 cells transfected with Aβ-CRY2-mCh where Aβ clusters formed upon stimulation with blue light (*Figure 7a*). The number of clusters per cell was greatly reduced by the addition of Li⁺ or the specific GSK3 inhibitor CHIR99021, but CHIR99021 additionally reduced the signal intensity of the clusters (*Figure 7b–c*, quantified 7d). These data demonstrated the potential use of the optogenetics system for drug testing.

Finally, we also extended these findings to a vertebrate system. We generated a zebrafish permanent line with UAS:Aβ-CRY2-mCh in the transgenic *TgBAC(gng8:GAL4)$^{c416}$* background (*Hong et al., 2013*) that limited expression to a few cells. We observed a diffuse mCh fluorescence in a small subset of neurons in the olfactory epithelium, the interpeduncular nucleus and in a few neurons sparsely distributed in the forebrain at 5 dpf. Upon blue light exposure (488 nM) the detectable intensity of the fluorescence increased rapidly (*Figure 8—figure supplement 1*). Several neurons among those expressing appeared to bleb and began to die within 40 to 50 min of initial exposure to the blue light (*Video 4*). Such damage was not observable in matched controls expressing Aβ-mCh driven by a CMV promotor lacking the CRY2 protein (*Video 5*). Our approach leads to the expression of Aβ-CRY2-mCh in only a small subset of neurons making analysis of AD hallmarks difficult, so we used transient expression of Aβ-CRY2-mCh driven by a ubiquitin promoter in 48 hr old embryos to quantify the effects of light-induced Aβ oligomerization on mitochondria health and metabolic deficits. Similar to the impairments observed in *C. elegans,* Aβ-CRY2-mCh expressing embryos showed lower maximum and spare respiration capacity compared to un-injected control (*Figure 8*). In addition, light exposure of Aβ-CRY2-mCh expressing embryos caused a significant reduction in ATP levels (*Figure 8a*).

## Discussion

Aggregation of Aβ has been studied extensively, but our results represent the first study to demonstrate that induced oligomerization of intracellular Aβ can be used to separate different pathologies (induced by Aβ oligomerization vs expression alone). A second approach to making inducible aggregates using a chemically controlled fluorescent protein has been published, but not yet widely tested (*Miyazaki et al., 2016*). Other previous approaches lacked an inducible oligomerization tool and could only demonstrate Aβ oligomers' toxicity through exogenous injection of Aβ oligomers (*Mroczko et al., 2018*). We use a tool to investigate the pathological effects of intracellular Aβ oligomerization in nematodes and flies, human kidney cells, and in the vertebrate model organism zebrafish (*Figure 8—figure supplement 1*, *Video 4*). Despite the reversibility of CRY2 clustering in previous findings, we find that CRY2 initiated clustering of Aβ appeared to be irreversible likely due

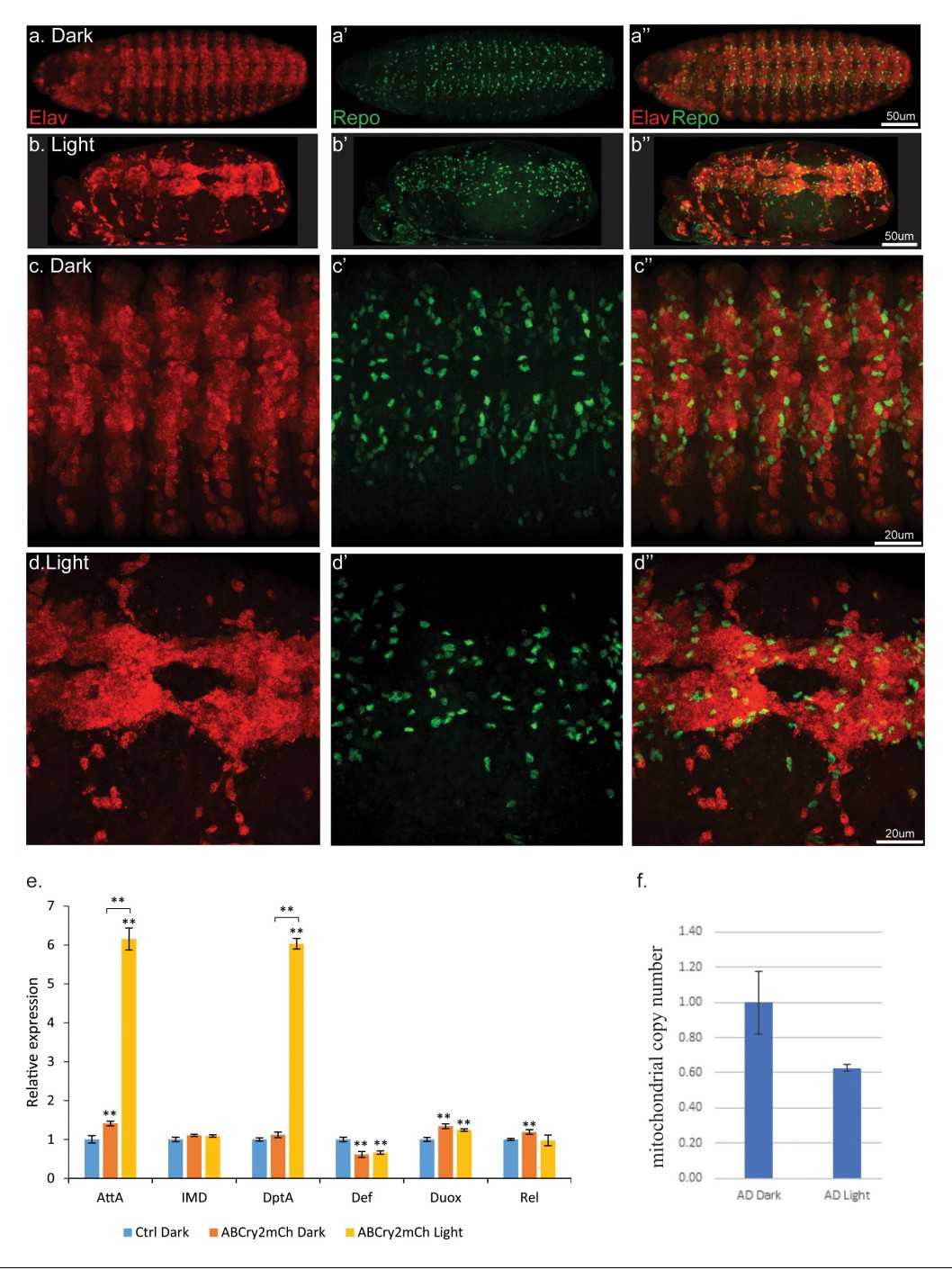

**Figure 4.** Light induced aggregation causes physical disruption of embryonic neural tissues. (**a–a''**) Elav-Gal4/UAS-Aβ-CRY2-mCh embryos kept in the dark and imaged for neurons (Elav) and glial cells (Repo). (**b–b''**) Elav-Gal4/UAS-Aβ-CRY2-mCh embryos exposed to light and imaged for neurons (Elav) and glial cells (Repo). (**c–d''**) Close ups. (**E**) Aggregation induced inflammatory response as measured by qPCR for inflammatory response genes. (**f**) The number of mitochondria in transgenic Aβ *Drosophila* embryo was reduced by 40% in light.

The online version of this article includes the following figure supplement(s) for figure 4:

**Figure supplement 1.** Light-induced Aβ aggregation leads to developmental deficits in transgenic *Drosophila* embryo.

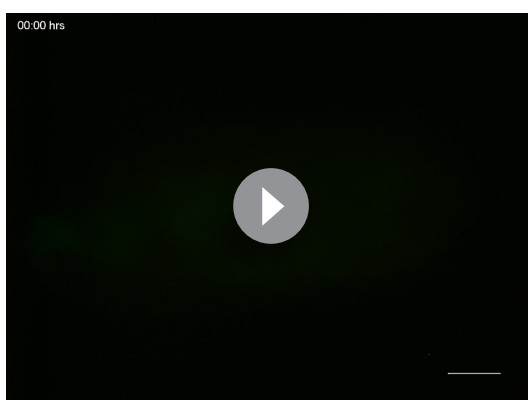

**Video 3.** Elav-Gal4/UAS-MyrTomato embryos exposed to light and imaged for neurons in red (tdTom) and glial cells Repo-QF2 >QUAS GFP. Blue laser power at higher setting to allow imaging of glial cells and control for laser damage.

https://elifesciences.org/articles/52589#video3

to the structure of the aggregates (*Antzutkin et al., 2012*). Consistent with existing literature showing that Aβ oligomers induce oxidative stress and inflammation (*Butterfield et al., 2013*; *Forloni and Balducci, 2018*), we show that light-induced oligomerization of Aβ, but not Aβ expression alone, is necessary for the metabolic defects, loss of mitochondria and inflammation in *C. elegans* and *Drosophila*. Light-induced oligomerization also leads to physical damage of the nervous system in *D. rerio*, resembling the loss of brain tissue in AD (*Figure 8—figure supplement 1*). The embryonic assay for physical damage phenocopies brain lesions only to some degree as the extent of the damage is enhanced by the forces applied to the developing neural tissue. The irreversibility of Aβ aggregates once initiated as seen here, also hints at intrinsic properties that may be unique or unusual to this peptide and warrants further investigation. It is also likely to prove a good model for testing anti-aggregation compounds.

We anticipate that our approach will serve as an attractive tool for carrying out drug screens and mechanistic studies for the treatment of AD. This optogenetic strategy will also undoubtedly complement other techniques due to the high level of spatiotemporal specificity. For example, it could be optimally positioned to gain insights into the unresolved mechanism of Aβ - whether oligomerization and subsequent accumulation or lack of effective clearance underpins AD pathologies. The separation of two phases of AD progression also suggests that single drug treatments will not suffice, and perhaps combinatorial approaches should be tried.

## Materials and methods

### Molecular cloning of optogenetic transgenes

DNA sequences corresponding to the human Amyloidβ1–42 amino acid sequence were synthesized (IDT) into a Gateway technology (Invitrogen) entry vector. The human Aβ1–42 sequence was used for *Drosophila* and zebrafish due to close evolutionary conservation and a nematode-codon-optimized version of Aβ1–42 was used for *C. elegans* (*Fong et al., 2016*). The CRY2-mCh fragment was cloned into pDONR vector with attB5 and attB2 sites, and MultiSite Gateway Pro 2.0 recombination (Invitrogen) was used to recombine donor plasmids and pDONR-CRY2-mCh into the respective destination vectors for three model organisms: a) pUASg.attB.3XHA for *Drosophila* to synthesize pUAS-Aβ-CRY2-mCh (*Bischof et al., 2007*) b) pDEST-hsp-16–2 p, a kind gift from Hidehito Kuroyanagi (Medical Research Institute, Tokyo Medical and Dental University) to synthesize hsp-16–2 p-Aβ-CRY2-mCh for *C. elegans* (*Okazaki et al., 2012*). c) pDEST-Tol2-PA2 (Invitrogen) and p5'E-Ubiquitin as described in *Kibat et al. (2016)* were used to make Ubiquitin driven pDEST-Tol2-PA2-Ubi-Aβ-CRY2-mCh and pDEST-Tol2-PA2 (Invitrogen) was used to make pDEST-Tol2-PA2-10xUAS-Aβ-CRY2-mCh for expression in *Danio rerio*. The CRY2 sequence were removed from this plasmid and the 10xUAS promoter were replaced with CMV promoter to produce pDEST-Tol2-PA2-CMV- Aβ-mCh by Gibson Assembly (NEB) for expression in *D. Rerio*.

d)pDEST40 (Invitrogen) was used to make CMV driven pDEST40-Aβ-CRY2-mCh for expression in HEK293 tissue culture cells.

### Crosses and expression of UAS construct

For *Drosophila*, the transgenes were injected into attP2 (Strain#8622) P[CaryP]attP268A4 by BestGene Inc (California) (*Groth et al., 2004*; *Markstein et al., 2008*). Expression was driven by Elav-GAL4 the neuronal driver and two ubiquitous drivers armadillo-GAL4 and daughterless-GAL4

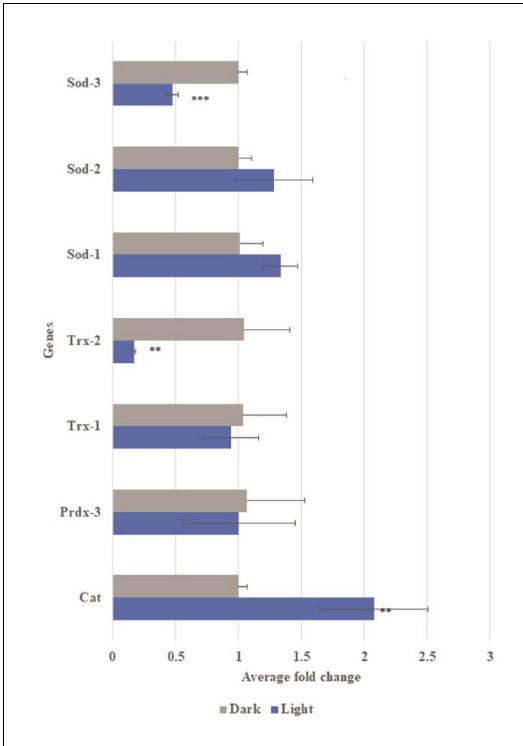

**Figure 5.** Gene expression of oxidative stress related genes in transgenic Aβ *C. elegans* with and without light-induction. Differential gene expression of catalase (Cat), peroxiredoxin-3 (Prdx3), thiroredoxin-1 (Trx-1), thioredoxin-2 (Trx-2), superoxide dismutase-1 (SOD-1), superoxide dismutase-2 (SOD-2) and superoxide dismutase-3 (SOD-3) in transgenic Aβ *C. elegans* in light condition compared to control dark condition. Light-induced Aβ aggregation showed up-regulation in expression of Cat (**p<0.01) and down-regulation in expression of Trx-2 (*p<0.01) and Sod-3 (***p<0.001).

(*Brand and Perrimon, 1993*). All additional stocks were obtained from the Bloomington *Drosophila* Stock Center (NIH P40OD018537) were used in this study.

Fly crosses performed were:

- Elav-Gal4 x w; UAS-Aβ-CRY2-mCh
- Arm-Gal4; Da-Gal4 x w; UAS-Aβ-CRY2-mCh
- Elav-Gal4; Repo-QF2, QUAS-GFP x UAS-Aβ-CRY2-mCh
- Elav-Gal4; Repo-QF2, QUAS-GFP x UAS-tdTomato
- esg-Gal4, UAS-GFP x w; UAS-Aβ-CRY2-mCh
- esg-Gal4, UAS-GFP x w; UAS-Arm-CRY2-mCh

For *C. elegans*, 50 ng/µl the construct hsp-16–2 p-Aβ-CRY2-mCh was co-injected with 25 ng/µl of pharyngeal-specific fluorescence marker *myo-2-gfp* into the distal gonads of wild type young adults. Transgenes were maintained as extrachromosomal arrays, hence careful selection of transgenic animals with pharyngeal GFP expression is required prior to all experiments. A *myo-2-gfp* strain *C. elegans* was also generated and used alongside as controls.

For *Danio rerio*, 50 ng/µl of the construct pDEST-Tol2-PA2-10xUAS-Aβ-CRY2-mCh and 60 ng/ µl Tol2RNA transposase were injected into TgBAC(gng8:GAL4) [c416] zebrafish embryos during the one-cell stage. pDEST-Tol2-PA2-Ubi-Aβ-CRY2-mCh and pDEST-Tol2-PA2-CMV-Aβ-mCh were injected into one-cell stage mitfa -/- zebrafish embryos lacking melanin pigment separately.

## Animal husbandry

*Drosophila* were maintained at standard humidity and temperature (25°C) with food containing 6 g Bacto agar, 114 g glucose, 56 g cornmeal, 25 g Brewer's yeast and 20 ml of 10% Nipagin in 1L final volume. Transgenic *Drosophila* and controls were distributed into either dark or light condition on Day 1. Transgenic flies fed with a lithium-supplemented diet were maintained in the same food as above with the addition of lithium chloride to a final concentration of 5 mM. *C. elegans* were maintained as previously described at 20°C (*Stiernagle, 2006*). Age-synchronized nematodes were generated by hypochlorite bleaching. 250 µM of 5-fluoro-2'-deoxyuridine (FUDR) was added to prevent progeny production in all experiments except for fertility assays. For Fertility assay, normal Nematode Growth Media (NGM) agar plates were used. To induce expression of Aβ-CRY2-mCh, Day 4 young adult worms were heat shocked at 35°C for 90 min, and subsequently incubated at 20°C for a day (Day 5) for recovery. The heat-shocked transgenic Day 6 *C. elegans* were then separated into dark or light condition and was maintained at 20°C throughout the experimental period. Non-heat shock controls were used for lifespan, fertility and locomotion studies. Adult zebrafish were reared under standard zebrafish facility conditions with a 14 hr light/10 hr dark cycle. Zebrafish embryos injected with different expression vectors were screened for fluorescence and distributed into either dark or light condition 24 hr post fertilization. Embryos were dechorionated at 48 hr post fertilization and exposed to blue light (488 nM) for 1–2 hr at room temperature to induce oligomerization and subsequently used for mitochondria metabolic flux assay or ATP assay.

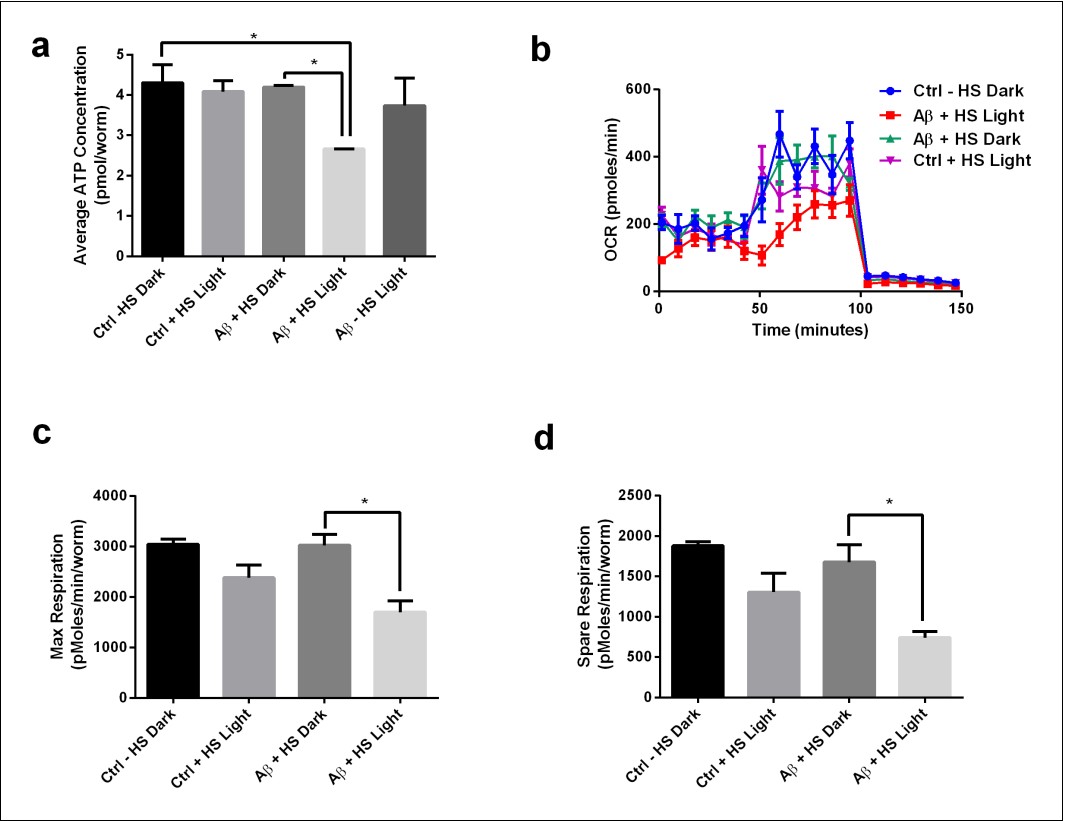

**Figure 6.** Light-induced Aβ oligomerization is required for the manifestation of metabolic defects in transgenic *C. elegans*. (a) ATP levels of control and transgenic Aβ nematodes (n = 3 repeats for each condition, with 100 nematodes per repeat; ANOVA post-test p<0.05, *). (b) Oxygen consumption profile (OCR) of control and transgenic Aβ nematodes (n = 6 repeats for each condition, with 10 nematodes per repeat). (c) Maximal respiration derived from OCR (n = 6 repeats for each condition, with 10 nematodes per repeat; ANOVA post-test p<0.05, *). (d) Spare respiration derived from OCR (n=6 repeats for each condition, with 10 nematodes per repeat; ANOVA post-test p<0.05, *).

## FRAP assay

Adult fly midguts were dissected in 200 µl of 1x PBS in a PYREX Spot Plates concave glass dish (FisherScientific). Subsequently, the guts were carefully transferred onto a small droplet of 1x PBS on a 35 mm glass bottom dish. Using fine forceps, the gut was repositioned to resemble its natural orientation. PBS was then removed from the area surrounding the gut, leaving a small amount of excess PBS to hold the gut in place and prevent desiccation. The 3 mm glass bottom dish was then mounted onto the Zeiss LSM800 (Carl Zeiss AG, Germany) for imaging. The super-resolution imaging function on the Zeiss LSM800 was used for the bleaching and fluorescence recovery. A small square within a cell was defined as the bleaching area and this same square was used for all bleaching experiments. Bleaching was performed at 4% laser power using the 488 nm laser for 1 s and the recovery was followed every 30 s for 20 minutes. The fluorescence intensity of the bleaching square ($F_S$), the entire cell ($F_C$) and the background ($F_B$) at each time point was computed by the ZEN 3.1 (Blue edition) software (Carl Zeiss, Germany). For determination of percentage fluorescence recovery, the background fluorescence was subtracted from the fluorescence of the bleaching square ($F_S$-$F_B$) and the fluorescence of the entire cell ($F_C$-$F_B$). The ($F_S$-$F_B$)/ ($F_C$-$F_B$) was computed at each time point and the ($F_S$-$F_B$)/($F_C$-$F_B$) before bleaching was set to 100%. This was done for all FRAP assays for both constructs and 3 separate experiments were conducted per construct.

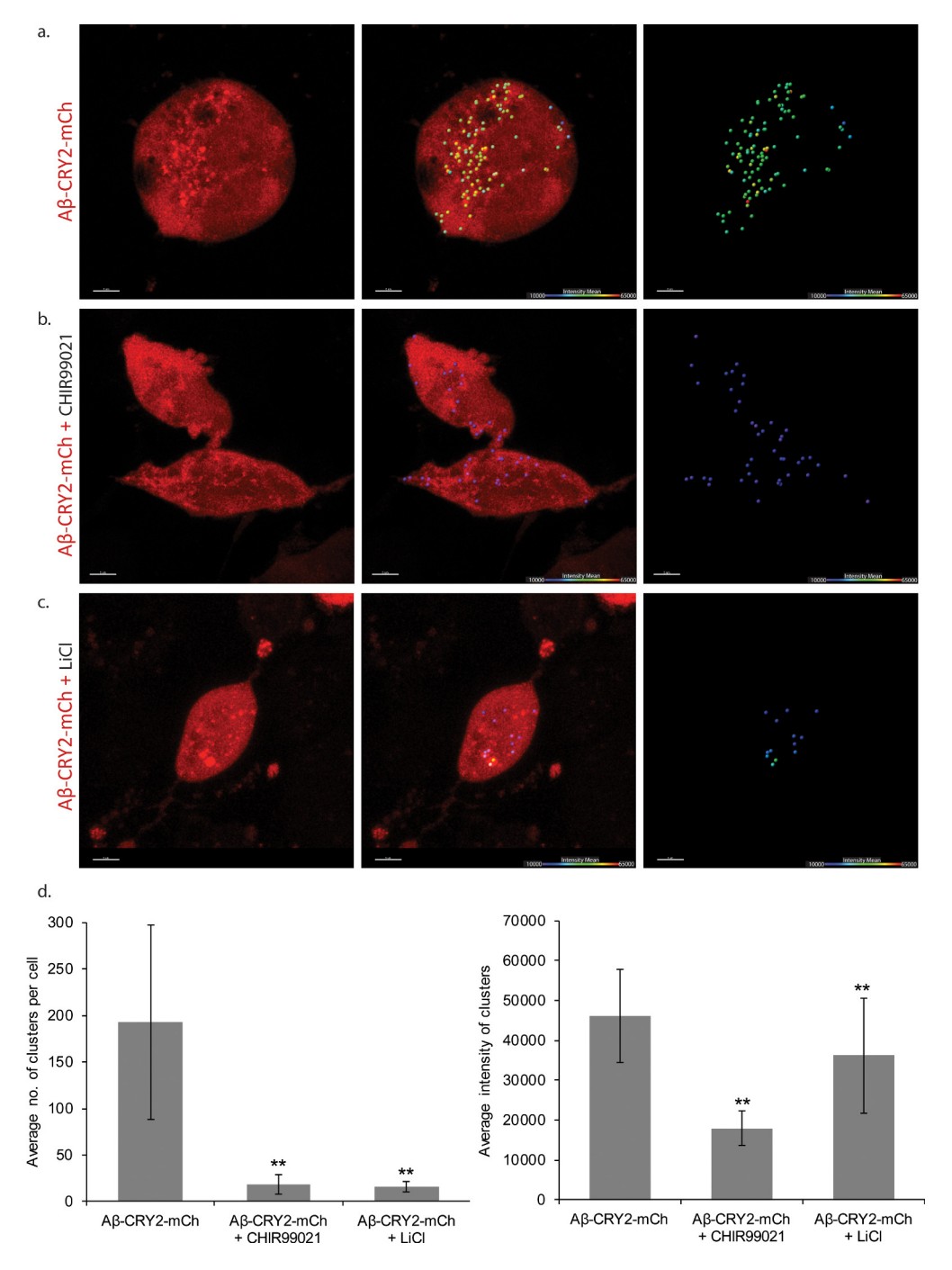

**Figure 7.** HEK cells expressing Aβ-CRY2-mCh in light condition. (**a**) Untreated control HEK cells show light-induced clustering of Aβ-CRY2-mCh and increased number of vacuoles under blue light illumination. (**b**) HEK cells treated with GSK3 inhibitor CHIR99021 (positive control) show little clustering of the light-induced Aβ-CRY2-mCh. Fewer cellular vacuoles observed. (**c**) HEK cells treated with lithium chloride show homogenous expression of Aβ-CRY2-mCh, whereby blue light illumination did not result in significant clustering of Aβ-CRY2-mCh. (**d**) Quantification of cluster number per cell showed significantly reduced cluster formation in cells treated with CHIR99021 and lithium chloride as compared to control cells. (**p<0.01) (**e**) Quantification of the mean intensity of the clusters showed significantly reduced intensity of the clusters when the cells were treated with GSK3 inhibitor CHIR99021 and lithium chloride as compared to control cells.

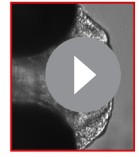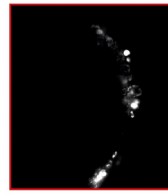

**Video 4.** Expression of Aβ-CRY2-mCh in zebrafish developing brains. Schematic of imaging set up and area of interest. Imaging of cell movement, Aβ-CRY2-mCh aggregation and cell death.
https://elifesciences.org/articles/52589#video4

## Lifespan assays

*Drosophila* and *C. elegans* were counted daily for the number of dead subjects and the number of censored subjects (excluded from the study). *Drosophila* that failed to respond to taps were scored as dead and those stuck to the food were censored. *C. elegans* that failed to respond to plate-tapping were scored as dead, those that burrowed to the side of the plate were censored.

## Locomotion assay

*Drosophila* locomotion was assessed using an established geotaxis assay previous described in *Rival et al. (2009)*. In brief, 25 *Drosophila* were enclosed in a plastic column (25 cm tall, with a 1.5 cm internal diameter) and were tapped to the bottom. The number of *Drosophila* at the top (Ntop) of the column, and that at the bottom (Nbot) were counted after 20 s. Three trials with the same sample were performed within 30 s interval. Performance index was defined as (15+Ntop-Nbot)/30.

Heat-shocked *C. elegans* were assessed at Day 9, by placing worms onto individual NGM plates, pre-spotted with *Escherichia coli* OP50. *C. elegans* were placed on the side of the bacteria spot on the NGM plate using a platinum worm picker and were left undisturbed to move freely for 15 min. Distance travelled was determined by imaging and measuring of worm tracks on bacteria using a stereomicroscope.

## Fertility assay

25 pairs of male and female *Drosophila* expressing MD-Aβ-Cry2-mCh were kept in a single tube under dark or light conditions. The number of eggs laid were scored after 5 hr 30 min for Days 5, 6, 7, 13, 14 and 20. Day 5 heat-shocked *C. elegans* were transferred onto individual NGM plates spotted with *E. coli*. Each worm was transferred to a fresh plate daily from Day 6 to Day 8, allowing 24 hr' egg laying period. The number of new young worms were counted on each plate as the progeny of a single individual.

## Adenosine triphosphate (ATP) assay

ATP assay using firefly lantern extract (Sigma-Aldrich) was performed as described by *Tsujimoto et al. (1970)*. Each *C. elegans* sample containing 100 D2 adult worms was freeze-thawed in liquid nitrogen and sonicated in 10% Trichloroacetic acid (TCA) buffer while each zebrafish sample containing five 48hpf embryos was lysed in 10% TCA buffer. The extracted ATP from the different conditions and ATP standards were loaded onto a 96-well plate and injected with firefly lantern extract. Luminescence was measured using a Cytation 3 Imaging Reader (Biotek).

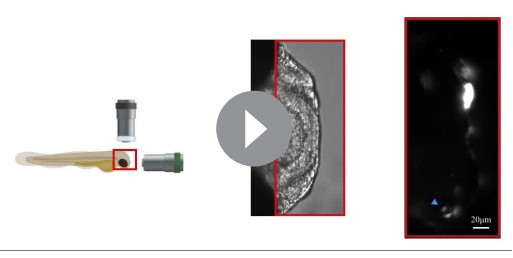

**Video 5.** Expression Aβ-mCh in zebrafish developing brains. Schematic of imaging set up and area of interest. No obvious cell movement or cell death despite Aβ-mCh expression.
https://elifesciences.org/articles/52589#video5

## Mitochondrial metabolic flux assay

Mitochondrial metabolic flux assay for *C. elegans* was performed as described by *Fong et al. (2017)* using XF96 Extracellular Flux Analyzer. 60 D2 adult worms from each condition 24 hr post light-treatment were transferred to a 96-well Seahorse plate containing M9 buffer. Final concentrations of the drugs used in the OCR experiment were 10 μM FCCP and 50 mM sodium azide. For zebrafish embryos, mitochondrial metabolic flux assay was performed as described by *Stackley et al. (2011)* with some modifications. Oxygen consumption rate (OCR) measurements were performed using the XF96 Extracellular

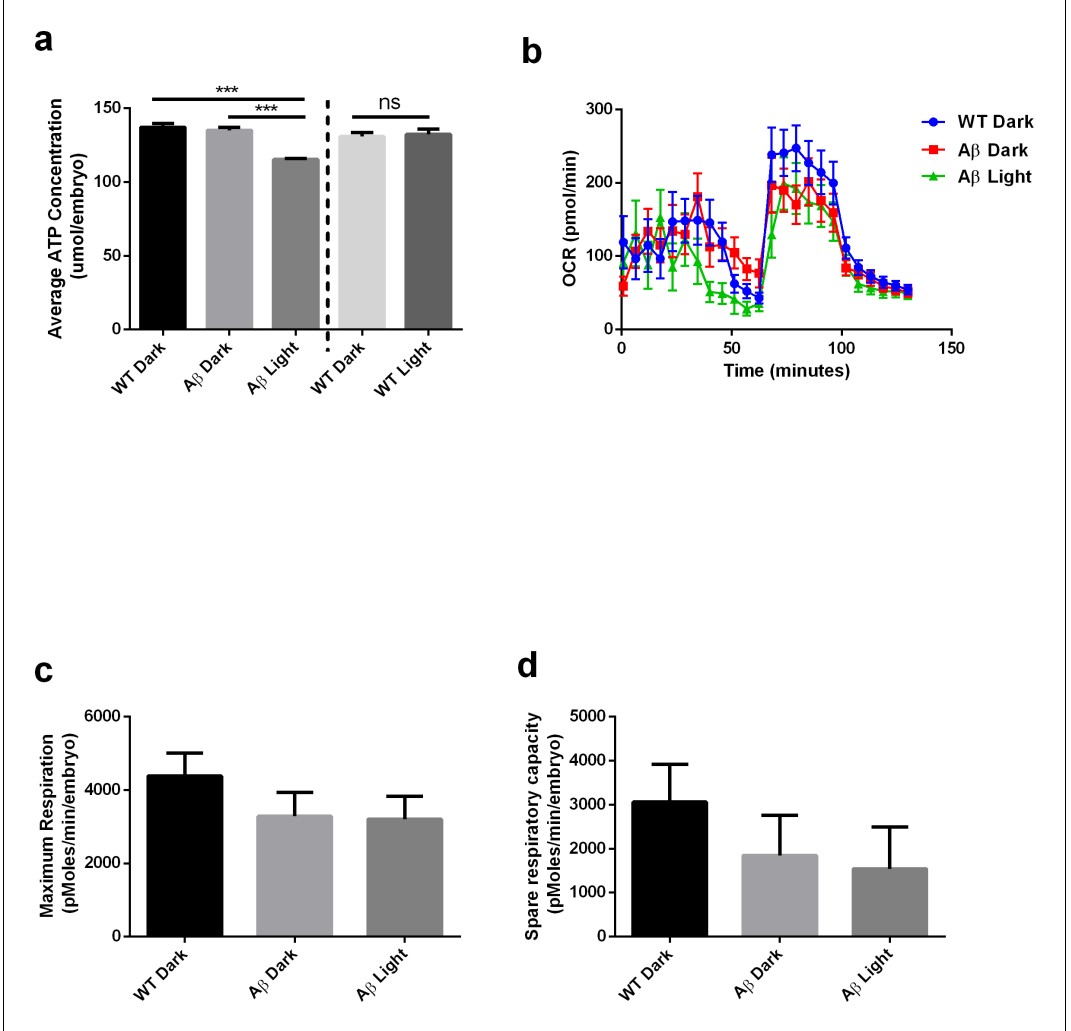

**Figure 8.** Light-induced Aβ oligomerization is required for the manifestation of metabolic defects in transgenic *D. Rerio*. (a) ATP levels of control and transgenic Aβ *D. rerio* embryos (n = 8 repeats for each condition; ANOVA post-test p<0.001, ***). Light treatment did not affect ATP levels in control embryos (performed in a separate experiment). (b) Oxygen consumption profile (OCR) of control and transgenic Aβ *D. rerio* embryos (24 embryos/condition) (c) Maximal respiration derived from OCR. (d) Spare respiration derived from OCR.

The online version of this article includes the following figure supplement(s) for figure 8:

**Figure supplement 1.** Transgenic Aβ model in *Danio Rerio embryo* expressing Aβ-CRY2-mCh (red).

Flux Analyzer. Each well contained one embryo (48hpf) in 175 µL of E3 medium (fish system water). Final concentrations of the drugs used in the OCR experiment were 9.4 µM oligomycin, 2.5 µM FCCP and 20 mM sodium azide.

## Light-sheet microscopy

*Drosophila* embryos were dechorionated using bleach, rinsed twice with water and dried, and loaded into a capillary filled with 1% low-melting agarose Type VII-A in water (Sigma) (*Colosimo and Tolwinski, 2006*). *C. elegans* and *D. rerio* were embedded directly in the agarose. Samples were imaged with a Lightsheet Z.1 microscope. Cry2 was activated by a dual-side illumination with 10% power 488 nm laser for 29.95 ms for every 2.5 min for 500 cycles. Controls were only exposed to 561 nm. Images were acquired with a water immersion objective at 10x/0.2 Illumination Optics and W Plan-Apochromat 20x.1.0 UV-VIS detection objective (Carl Zeiss, Germany). Image data were

processed using the maximum intensity projection function of ZEN 2014 SP software (Carl Zeiss, Germany), and were analyzed with ImageJ (NIH) and IMARIS 9.0 (Bitplane AG, UK).

## Transfection and live cell confocal microscopy

Human Embryonic Kidney (HEK293T) cells were obtained from ATCC through the local distributor Bio-REV Pte Ltd after Short Tandem Repeat profiling, confirmed mycoplasma-free and maintained in Dulbecco's Modified Eagle's Medium with 10% Fetal Bovine Serum and 1% Penicillin and Streptomycin (Invitrogen). Cells were plated at 80% confluence on 35 mm TC treated glass bottom dish 24 hr prior to transfection with 2.5 mg of pDEST40-Aβ-CRY2-mCh using the Effectene Transfection Reagent (Qiagen). A transfection master mix containing the plasmid and transfection reagent was prepared for all treatment plates during each biological replicate to ensure the transfection efficiency was the same across all plates. Two hours after transfection, the control plate was topped up with 250 µl of complete media without antibiotics. For cells treated with drugs, LiCl or CHIR99021 were also added at this point to achieve a final concentration of 2 mM LiCl or 3 mM CHIR99021 respectively. Cells transfected with only the pDEST40-Aβ-CRY2-mCh plasmid were used as negative control. Cells were imaged 24 hr after transfection using the Zeiss LSM800 confocal microscope with 63x oil immersion lens.

## Immunofluorescence and confocal microscopy

*Drosophila* embryos 9 hr after deposition were incubated at 25°C in either light or dark conditions for 13 hr before being dechorionated using bleach. Embryos were then fixed with Heat-Methanol treatment (*Müller and Wieschaus, 1996*) or with heptane/4% formaldehyde in phosphate buffer (0.1M NaPO4 pH 7.4) (*Tolwinski and Wieschaus, 2001*). Staining, detection and image processing as described in *Colosimo and Tolwinski (2006)*.

Primary antibodies used were the glial cell marker anti-Repo (mouse mAb, 8D12) and the neuronal cell marker anti-Elav (rat mAb, 7EA810) from Developmental Studies Hybridoma Bank (DSHB developed under the auspices of the NICHD and maintained by The University of Iowa, Department of Biological Sciences, Iowa City, IA 52242). Secondary antibodies used were Alexa Flour 488 anti-rat and Alexa Flour 647 anti-mouse (Invitrogen).

For detection of Aβ aggregates, thioflavin T (ThT) staining were performed as previously described *Iijima et al. (2008)*. Formaldehyde-fixed embryos were incubated in 50% EtOH containing 0.1% ThT (Sigma) overnight. Embryos were destained in 50% EtOH for 10 min, followed by three washes in PBS. Embryos were then mounted on microscope slides using Aquapolymount (Polysciences, Inc).

Images were acquired on the Zeiss LSM 800 (Carl Zeiss, Germany) using the following settings: 1% laser power for 488 nm; 5% laser power for 561 nm; 2% laser power for 647 nm. Images were processed using the ZEN 2014 SP1 software (Carl Zeiss, Germany) and Imaris (Bitplane AG).

## RNA extraction, cDNA synthesis and qPCR

*Drosophila* embryos 9 hr after deposition were incubated at 25°C in either light or dark conditions for 13 hr were used. Embryos were dechorionated and washed with 100% ethanol prior to RNA extraction using the ISOLATE II RNA Mini Kit's protocol (Bioline, UK). The extracted RNA was quantified using Nanodrop (Thermo Fisher Scientific). cDNA synthesis was done according to the SensiFAST cDNA Synthesis Kit's protocol (Bioline). Primers pairs used: AttA, IMD, DptA, Def, Duox, Rel, catalase, Prdx3, Trx1, Trx2, SOD-1, SOD-2, SOD-3 and reference gene Rpl32. Quantitative PCR was performed using SYBR Green. Expression data were normalized to the dark controls. For mitochondrial copy number, the above collection method was used except the following: the embryos were stored in tritonX-100 and the primers used were Mitochondria Cytochrome b (MtCyb) and reference gene RNAse P.

## Image data and statistical analysis

Aβ aggregates were quantified in *Drosophila* using ImageJ (NIH) and in *C. elegans* and HEK cells with IMARIS. For HEK cells, the number of clusters larger than 0.4 µm in diameter were determined for at least 3 cells for the control and drug treated cells to calculate the average number of clusters per cell. The mean intensity per cluster across all these cells was also determined to calculate the

average intensity of the clusters for the control and drug treated cells. For statistical analysis of the expression of genes, student's t-test was used for except for cases where the data showed unequal standard deviation (F-test, p<0.05), in which the Mann-Whitney nonparametric test was performed. Statistical analysis of lifespan studies and behavioral assays were performed using OASIS 2 (*Han et al., 2016*). The number of samples was determined empirically. All graphs were plotted using Graphpad PRISM 6 (Graphpad Software).

## Key resources table

New model organisms generated for this paper are available to the community by contacting the authors. Stable lines and stocks will be made available through stock centers.

*Drosophila* line UAS-Aβ-Cry2-mCh inserted on Chromosome 3 site attP2:

- *D. rerio* UAS-Aβ-CRY2-mCh; *TgBAC(gng8:GAL4)* $^{c416}$
- *C. elegans* unintegrated line hsp-16–2 p-Aβ-CRY2-mCh.

## Acknowledgements

We acknowledge Lavonna Mark and Rebecca Rubright for assistance with initial experiments using the zebrafish. We are thankful for the funding provided by Ministry of Education Singapore AcRF grant IG17-BS101 to JG, Yale-NUS College grant R-607-265-225-121 to ASM, AcRF grants IG17-LR006 and IG18-LR001 to NST.

## Additional information

### Funding

| Funder | Grant reference number | Author |
| --- | --- | --- |
| Ministry of Education - Singapore | IG17-LR001 | Nicholas S Tolwinski |
| Ministry of Education - Singapore | IG17-BS101 | Jan Gruber |
| Ministry of Education - Singapore | IG18-LR001 | Nicholas S Tolwinski |
| Yale-NUS College | R-607-265-225-121 | Ajay S Mathuru |

The funders had no role in study design, data collection and interpretation, or the decision to submit the work for publication.

### Author contributions

Chu Hsien Lim, Conceptualization, Investigation, Methodology; Prameet Kaur, Data curation, Investigation, Methodology; Emelyne Teo, Data curation, Validation, Investigation, Methodology; Vanessa Yuk Man Lam, Caroline Kibat, Investigation, Methodology; Fangchen Zhu, Investigation, Writing - original draft, Writing - review and editing; Jan Gruber, Resources, Supervision, Funding acquisition, Methodology; Ajay S Mathuru, Resources, Data curation, Supervision, Investigation, Methodology; Nicholas S Tolwinski, Conceptualization, Data curation, Formal analysis, Supervision, Funding acquisition, Investigation, Methodology

### Author ORCIDs

Chu Hsien Lim (iD) https://orcid.org/0000-0001-6691-8277
Emelyne Teo (iD) https://orcid.org/0000-0001-5050-4109
Jan Gruber (iD) http://orcid.org/0000-0003-3329-3789
Ajay S Mathuru (iD) http://orcid.org/0000-0003-4591-5274
Nicholas S Tolwinski (iD) https://orcid.org/0000-0002-8507-2737

### Decision letter and Author response

Decision letter https://doi.org/10.7554/eLife.52589.sa1

Author response https://doi.org/10.7554/eLife.52589.sa2

## Additional files

### Supplementary files
• Transparent reporting form

### Data availability
All data generated or analysed during this study are included in the manuscript and supporting files.

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
