## [Decision Letter]

**Acceptance summary:**

This manuscript describes a new optogenetic system for studying the impact of amyloid-β oligomers in vivo. The authors describe light-induced oligomerization in each system and an initial characterization of phenotypes. Of note is the fact that the authors generated four separate models that fall into three distinct categories: in vitro (cell culture), invertebrate (*C. elegans, D. melanogaster*), and vertebrate (*D. rerio*) to corroborate their findings and to demonstrate their utility. That the authors took the time and effort to generate multiple models, which could be of use to a variety of researchers working in the area of neurodegeneration is commendable.

**Decision letter after peer review:**

Thank you for submitting your article "Application of Optogenetic Amyloid-β Distinguishes Between Metabolic and Physical Damage in Neurodegeneration" for consideration by *eLife*. Your article has been reviewed by three peer reviewers, and the evaluation has been overseen by a Reviewing Editor and Utpal Banerjee as the Senior Editor. The following individuals involved in review of your submission have agreed to reveal their identity: Michael Petrascheck (Reviewer #2).

The reviewers have discussed the reviews with one another and the Reviewing Editor has drafted this decision to help you prepare a revised submission.

Summary:

This manuscript describes a new optogenetic system for studying the impact of amyloid-β oligomers in vivo. They provide evidence for light-induced oligomerization in each system and an initial characterization of phenotypes. Of note, is the fact that the authors generated four separate models that fall into three distinct categories: in vitro (cell culture), invertebrate (*C. elegans, D. melanogaster*), and vertebrate (*D. rerio*) to corroborate their findings and to demonstrate their utility. That the authors took the time and effort to generate multiple models, which could be of use to a variety of researchers working in the area of neurodegeneration is commendable. However, some significant limitations and missing controls were identified in the current manuscript that reduce the utility of the system and the strength of the conclusion. These should be addressed prior to publication.

Essential revisions:

1) Key controls appear to be missing from several of the phenotypic characterization experiments. Non-transgenic animals under dark and light conditions should be shown for each experiment, so that any effects of the light treatment can be understood. It would also be useful to have control strains where amyloid-β-mCH is expressed without the CRY2 to demonstrate that the phenotypes are caused by the oligomerization rather than an interaction between the amyloid-β and/or mCH with light treatment.

2) We don't understand why the authors seem to assume that oligomerization does not also happen in the dark conditions. It has been shown that transgenically expressed amyloid-β in both worms and flies will aggregate in a time-dependent manner in the absence of the optogenetic system. Might these studies reflect a relative difference in the abundance of oligomers and aggregates rather than an all or none comparison. The text should be rewritten to reflect this or an explanation otherwise should be provided.

3) There is no evidence that the aggregates seen here are related to the aggregates found in neurodegenerative disease. Most of the aggregating protein consists of Cry and mCherry with only little Aβ. Some biochemistry or Fluorescence recovery after photobleaching (FRAP) experiments to characterize the light induced aggregates should be provided. Are these aggregates protease resistant?, Are the fluorescent aggregates really immobile or are they lose aggregates as determined by FRAP? The ThT stain in S4 suggest that they may even be amyloids. Especially given that the aggregates appear to be toxic, it would be really important to know what sort of aggregate is produced through light induced polymerization.

4) The authors hypothesize that Li^+^ rescues the lifespan deficit induced by AB oligomers via Wnt signaling, however, no evidence is provided other than a reference to previously published studies. It would be good to know that Wnt signaling is indeed the mechanism in this case, and in addition to measure the physical and metabolic damage that the authors report in the first part of their study for at least one additional model rather than mixing and matching between *C. elegans* and *D. melanogaster* (ideally, one of the invertebrate models and the vertebrate model). Because this is the first in vivo report of its kind using optogenetics, it would be preferable to show that what is expected (i.e. Li^+^ extending lifespan and curbing detrimental effects of AB oligomers) actually occurs via the expected/previously described mechanism.

5) Why were the physiology and/or health status parameters such as sensorimotor function, ATP deficit, number of mitochondria and oxidative damage not measured in the only vertebrate model? In addition to neuronal death, these endpoints would make a more convincing case for this model.

6) Information about replication for many of the phenotypic assays appears to be missing and should be provided. Figure 1 needs better quantification, with regards to number of aggregates as well as intensity to conduct statistical tests to determine real differences. Figure 7 needs quantification, with regards to number of aggregates as well as intensity of them in HEK293 cells. With quantitation this could be a very strong piece of data.

7) The use of the hsp-16 promoter in *C. elegans* is unfortunate as the need for heat shock creates a confound. Induction of chaperones from the heat shock itself is likely to impact the dynamics of amyloid-β, as has been previously shown in other transgenic lines. Optimally, the authors would create another inducible or constitutive system which could be compared to the heat shock system in order to understand the impact of chaperone induction. At a minimum a detailed description of this limitation and a justification for why the authors do not feel this is a significant limitation should be provided.

---

## [Author Response]

Essential revisions:1) Key controls appear to be missing from several of the phenotypic characterization experiments. Non-transgenic animals under dark and light conditions should be shown for each experiment, so that any effects of the light treatment can be understood. It would also be useful to have control strains where amyloid-β-mCH is expressed without the CRY2 to demonstrate that the phenotypes are caused by the oligomerization rather than an interaction between the amyloid-β and/or mCH with light treatment.

We thank the reviewers for pointing out these limitations. We did not perform all of the specific controls for CRY2 as the CRY2 system has been extensively studied in a variety of systems, for examples please see [1-12]. The discovery of the oligomerizing domain [13], led to its adoption for study of protein-protein interactions [14] and signaling in mammalian cells [2] and introduction into insect development [4, 10, 15], transcription [6], and signaling studies [8, 9, 12, 16]. Therefore, the comparison we undertook was mainly based on light conditions.

We have attempted to address the concerns by adding a control removing CRY2 from the construct making Aβ-mCherry expressed in zebrafish. This construct does not cause neuronal cell death, Video 5.

We have added controls for *C. elegans* experiments for light and dark and heat shock, see below. For *Drosophila*, we cannot complete the additional controls in the two month window as it requires generating new lines expressing Aβ-mCherry, CRY2-mCherry, CRY2 and mCherry alone from the att40 site we used for the Aβ-CRY2-mCherry insertion.

2) We don't understand why the authors seem to assume that oligomerization does not also happen in the dark conditions. It has been shown that transgenically expressed amyloid-β in both worms and flies will aggregate in a time-dependent manner in the absence of the optogenetic system. Might these studies reflect a relative difference in the abundance of oligomers and aggregates rather than an all or none comparison. The text should be rewritten to reflect this or an explanation otherwise should be provided.

We apologize for this mistake. The text has been changed to reflect this more clearly. We did not intend to imply that oligomerization does not occur, just that the optogenetic approach increases the rate.

3) There is no evidence that the aggregates seen here are related to the aggregates found in neurodegenerative disease. Most of the aggregating protein consists of Cry and mCherry with only little Aβ. Some biochemistry or Fluorescence recovery after photobleaching (FRAP) experiments to characterize the light induced aggregates should be provided. Are these aggregates protease resistant?, Are the fluorescent aggregates really immobile or are they loose aggregates as determined by FRAP? The ThT stain in S4 suggest that they may even be amyloids. Especially given that the aggregates appear to be toxic, it would be really important to know what sort of aggregate is produced through light induced polymerization.

As the reviewers point out, the only evidence that we present that matches the human disease aggregates is the ThT staining. In this paper, we focus on intracellular oligomerization, comparing fast and slow, relating to our previous work on intracellular Aβ [17, 18].

We have added an experiment tracking a closeup of an aggregate. We have added the FRAP experiment looking at the dynamics of clusters and comparing it to an unrelated UAS-Arm-CRY2-mCherry construct[9].

The clusters appear to be mobile within cells as can be seen in the zebrafish video especially. They are much less dynamic than the comparable Arm construct.

4) The authors hypothesize that Li^+^ rescues the lifespan deficit induced by AB oligomers via Wnt signaling, however, no evidence is provided other than a reference to previously published studies. It would be good to know that Wnt signaling is indeed the mechanism in this case, and in addition to measure the physical and metabolic damage that the authors report in the first part of their study for at least one additional model rather than mixing and matching between *C. elegans* and *D. melanogaster* (ideally, one of the invertebrate models and the vertebrate model). Because this is the first in vivo report of its kind using optogenetics, it would be preferable to show that what is expected (i.e. Li^+^ extending lifespan and curbing detrimental effects of AB oligomers) actually occurs via the expected/previously described mechanism.

We agree that this was not well supported and presented more as a possible mechanism for Lithium. We have changed the text to reflect the various possible mechanisms by which Lithium has been proposed to function in AD. The tissue culture results point to GSK3 as the target as both Li^+^ and CHIR target it. A full analysis of Wnt isn’t feasible in the two month window, but is now underway.

We have added biochemical analysis of the vertebrate model zebrafish confirming the *C. elegans* findings.

5) Why were the physiology and/or health status parameters such as sensorimotor function, ATP deficit, number of mitochondria and oxidative damage not measured in the only vertebrate model? In addition to neuronal death, these endpoints would make a more convincing case for this model.

An excellent point, but the Zebrafish model presented only expressed the optogenetic construct in a subset of neurons making global analysis difficult as most cells were unaffected. We have now made fish transiently expressing the construct ubiquitously and measured several parameters. The new figure shows the same trends as in the *C. elegans* model.

6) Information about replication for many of the phenotypic assays appears to be missing and should be provided. Figure 1 needs better quantification, with regards to number of aggregates as well as intensity to conduct statistical tests to determine real differences. Figure 7 needs quantification, with regards to number of aggregates as well as intensity of them in HEK293 cells. With quantitation this could be a very strong piece of data.

In response, we have added N numbers to the phenotypic assays. We have added graphs to Figure 1 and quantified the aggregates in Figure 7.

7) The use of the hsp-16 promoter in *C. elegans* is unfortunate as the need for heat shock creates a confound. Induction of chaperones from the heat shock itself is likely to impact the dynamics of amyloid-β, as has been previously shown in other transgenic lines. Optimally, the authors would create another inducible or constitutive system which could be compared to the heat shock system in order to understand the impact of chaperone induction. At a minimum a detailed description of this limitation and a justification for why the authors do not feel this is a significant limitation should be provided.

We agree with this assessment. The choice was made based on the availability of Gateway ready vectors for *C. elegans* and new lines are being made in line with our recent AD model [18]. Since, as part of this study, we generated models in different model organisms in parallel, availability of reagents facilitating multiplexing was critical and this informed our choice. We agree that this imposes limitations and have therefore modified the main text of the manuscript to more fully reflect and discuss them.

To further understand how heat-shock treatment may impact the dynamics of Aβ in our transgenic strains and to separate the effects of heat-shock treatment from Aβ expression, we have included the following controls in all metabolic and phenotypic assays:

a) transgenic animals in both heat-shock (Aβ HS L) and non-heat-shock (Aβ -HS L)

b) non-transgenic control animals in both heat-shock (Ctrl HS) and non-heat-shock (Ctrl –HS)

As suggested by the reviewer, we have now added a detailed description of this limitation and a justification involving the comparison of the above controls in the text as follows:

“Induction of chaperones from heat-shock has previously been shown to impact the dynamics of Aβ. […] However, animals that had been heat-shocked had significantly higher Aβ levels compared to non-heat-shocked animals, suggesting that heat-shock, as intended, drove higher level of Aβ expression and that this was not compensated for by secondary induction of chaperones.”

“To further examine whether the metabolic detriments were mediated by heat-shock treatment, we have included a heat-shocked non-transgenic control in the metabolic and phenotypic assays. […] This suggests that regardless of the effects of heat-shock and chaperone induction, oligomerization still negatively impacts the phenotypes.”

**Reference:**

[1] Bugaj L, Spelke D, Mesuda C, Varedi M, Kane R, Schaffer D. Regulation of endogenous transmembrane receptors through optogenetic CRY2 clustering. Nature communications. 2015;6:6898.

[2] Bugaj LJ, Choksi AT, Mesuda CK, Kane RS, Schaffer DV. Optogenetic protein clustering and signaling activation in mammalian cells. Nat Methods. 2013;10:249-52.

[3] Bugaj LJ, O’donoghue GP, Lim WA. Interrogating cellular perception and decision making with optogenetic tools. J Cell Biol. 2017;216:25-8.

[4] Guglielmi G, Barry JD, Huber W, De Renzis S. An Optogenetic Method to Modulate Cell Contractility during Tissue Morphogenesis. Dev Cell. 2015;35:646-60.

[5] Herrera Perez M, Kasza K. Optogenetic control of contractile tissue forces during *Drosophila* morphogenesis. APS Meeting Abstracts2019.

[6] Huang A, Amourda C, Zhang S, Tolwinski NS, Saunders TE. Decoding temporal interpretation of the morphogen Bicoid in the early *Drosophila* embryo. eLife. 2017;6.

[7] Idevall-Hagren O, Dickson EJ, Hille B, Toomre DK, De Camilli P. Optogenetic control of phosphoinositide metabolism. Proc Natl Acad Sci U S A. 2012;109:E2316-23.

[8] Johnson HE, Goyal Y, Pannucci NL, Schupbach T, Shvartsman SY, Toettcher JE. The Spatiotemporal Limits of Developmental Erk Signaling. Dev Cell. 2017;40:185-92.

[9] Kaur P, Saunders TE, Tolwinski NS. Coupling optogenetics and light-sheet microscopy, a method to study Wnt signaling during embryogenesis. Sci Rep. 2017;7:16636.

[10] Krueger D, Izquierdo E, Viswanathan R, Hartmann J, Cartes CP, De Renzis S. Principles and applications of optogenetics in developmental biology. Development. 2019;146:dev175067.

[11] Gordley RM, Bugaj LJ, Lim WA. Modular engineering of cellular signaling proteins and networks. Current opinion in structural biology. 2016;39:106-14.

[12] Bugaj L, Sabnis A, Mitchell A, Garbarino J, Toettcher JE, Bivona T, et al. Cancer mutations and targeted drugs can disrupt dynamic signal encoding by the Ras-Erk pathway. Science. 2018;361:eaao3048.

[13] Más P, Devlin PF, Panda S, Kay SA. Functional interaction of phytochrome B and cryptochrome 2. Nature. 2000;408:207.

[14] Kennedy MJ, Hughes RM, Peteya LA, Schwartz JW, Ehlers MD, Tucker CL. Rapid blue-light-mediated induction of protein interactions in living cells. Nat Methods. 2010;7:973-5.

[15] Deneke VE, Puliafito A, Krueger D, Narla AV, De Simone A, Primo L, et al. Self-Organized Nuclear Positioning Synchronizes the Cell Cycle in *Drosophila* Embryos. Cell. 2019;177:925-41. e17.

[16] Johnson HE, Shvartsman SY, Toettcher JE. Optogenetic rescue of a developmental patterning mutant. BioRxiv. 2019:776120.

[17] Fong S, Teo E, Ng LF, Chen C-B, Lakshmanan LN, Tsoi SY, et al. Energy crisis precedes global metabolic failure in a novel *Caenorhabditis elegans* Alzheimer disease model. Scientific reports. 2016;6:33781.

[18] Teo E, Ravi S, Barardo D, Kim H-S, Fong S, Cazenave-Gassiot A, et al. Metabolic stress is a primary pathogenic event in transgenic *Caenorhabditis elegans* expressing pan-neuronal human amyloid beta. eLife. 2019;8.

[19] Wu Y, Cao Z, Klein WL, Luo Y. Heat shock treatment reduces beta amyloid toxicity in vivo by diminishing oligomers. Neurobiology of aging. 2010;31:1055-8.

[20] Fonte V, Kipp DR, Yerg J, Merin D, Forrestal M, Wagner E, et al. Suppression of in Vivo β-Amyloid Peptide Toxicity by Overexpression of the HSP-16.2 Small Chaperone Protein. Journal of Biological Chemistry. 2008;283:784-91.